# Evaluation of the Structural Modification of Ibuprofen on the Penetration Release of Ibuprofen from a Drug-in-Adhesive Matrix Type Transdermal Patch

**DOI:** 10.3390/ijms23147752

**Published:** 2022-07-13

**Authors:** Paula Ossowicz-Rupniewska, Paulina Bednarczyk, Małgorzata Nowak, Anna Nowak, Wiktoria Duchnik, Łukasz Kucharski, Joanna Klebeko, Ewelina Świątek, Karolina Bilska, Joanna Rokicka, Ewa Janus, Adam Klimowicz, Zbigniew Czech

**Affiliations:** 1Department of Chemical Organic Technology and Polymeric Materials, Faculty of Chemical Technology and Engineering, West Pomeranian University of Technology in Szczecin, Piastów Ave. 42, PL-71065 Szczecin, Poland; paulina.bednarczyk@zut.edu.pl (P.B.); nowak.malgorzata@zut.edu.pl (M.N.); joanna.klebeko@gmail.com (J.K.); ewelina.swiatek@zut.edu.pl (E.Ś.); karolinabilska99@gmail.com (K.B.); joanna.rokicka@zut.edu.pl (J.R.); ejanus@zut.edu.pl (E.J.); psa_czech@wp.pl (Z.C.); 2Department of Cosmetic and Pharmaceutical Chemistry, Pomeranian Medical University in Szczecin, Powstańców Wielkopolskich Ave. 72, PL-70111 Szczecin, Poland; anowak@pum.edu.pl (A.N.); wiktoria.duchnik@pum.edu.pl (W.D.); lukasz.kucharski@pum.edu.pl (Ł.K.); adam.klimowicz@pum.edu.pl (A.K.)

**Keywords:** structural modification of ibuprofen, acrylic pressure-sensitive adhesives, transdermal patch, shear strength, adhesion, tack

## Abstract

This study aimed to evaluate the effect of chemical modifications of the structure of active compounds on the skin permeation and accumulation of ibuprofen [IBU] from the acrylic pressure-sensitive adhesive used as a drug-in-adhesives matrix type transdermal patch. The active substances tested were ibuprofen salts obtained by pairing the ibuprofen anion with organic cations, such as amino acid isopropyl esters. The structural modification of ibuprofen tested were Ibuprofen sodium salt, [GlyOiPr][IBU], [AlaOiPr][IBU], [ValOiPr][IBU], [SerOiPr][IBU], [ThrOiPr][IBU], [(AspOiPr)_2_][IBU], [LysOiPr][IBU], [LysOiPr][IBU]_2_, [PheOiPr][IBU], and [ProOiPr][IBU]. For comparison, the penetration of unmodified ibuprofen and commercially available patches was also investigated. Thus, twelve transdermal patches with new drug modifications have been developed whose adhesive carrier is an acrylate copolymer. The obtained patches were characterized for their adhesive properties and tested for permeability of the active substance. Our results show that the obtained ibuprofen patches demonstrate similar permeability to commercial patches compared to those with structural modifications of ibuprofen. However, these modified patches show an increased drug permeability of 2.3 to even 6.4 times greater than unmodified ibuprofen. Increasing the permeability of the active substance and properties such as adhesion, cohesion, and tack make the obtained patches an excellent alternative to commercial patches containing ibuprofen.

## 1. Introduction

Ibuprofen (IBU), as a nonsteroidal anti-inflammatory drug (NSAID), is primarily and routinely administered orally and topically to ease moderate pain [1,2,3]. This drug is poorly soluble in aqueous media, and thus, the rate of dissolution from the currently available solid dosage forms is limited. This leads to poor bioavailability at high doses after oral administration, thereby increasing the risk of unwanted adverse effects, including stomach inflammation (gastritis) resulting in a stomach ulcer or even bleeding [1,3]. Poor solubility is a problem for developing injectable solution dosage forms. Because of its poor skin permeability, it is difficult to obtain an effective therapeutic concentration from topical preparations [1].

The transdermal drug delivery system (TDDS) is one of the essential methods of delivering the drug to the body, thus constituting an attractive route for drug delivery as well as a challenging area of research [3,4]. Topical delivery offers compelling advantages compared to more conventional delivery systems. It provides many advantages over conventional routes by enhancing patient compliance, avoiding first-pass metabolism, and minimizing harmful side effects from an overdose. Additionally, it can address the limited controlled release and the low bioavailability of many oral drugs, avoiding potential damage to the gastrointestinal tract. Transdermal delivery systems may also be preferred as they are non-invasive, self-administered, and generally inexpensive [3,5,6,7]. A primary disadvantage of TDDS is that they are frequently unable to convey the desired drug through the skin. Thus, only a limited number of drugs are amenable to administration by this route. With current delivery methods, successful transdermal drugs have molecular masses that are only up to a few hundred Daltons, exhibit octanol-water partition coefficients that heavily favor lipids, and require doses of milligrams per day or less, so it is difficult to exploit the transdermal route to deliver hydrophilic drugs [6,8]. The main barrier to efficient penetration is the complex structure of the epidermis which forces the active substance to move between many hydrophilic and hydrophobic domains. Therefore, it is ideal that a drug used in TDDS is both hydrophobic and hydrophilic or only lipophilic. In addition, it must be of low molecular weight and efficacious at low doses [6]. Therefore, TDDS is a required method, especially when using NSAIDs, including ibuprofen (IBU), characterized by a low molecular weight with appropriate physicochemical properties, such as lipophilic character, to be used transdermally [3,9,10].

Recent research on TDDS with ibuprofen focuses mainly on the use of a chemical modification of the structure of ibuprofen [4,11,12,13] and the use of permeation enhancers (such as short-chain alcohols or light mineral oil as a lipophilic vehicle) or crystallization inhibitors that will increase the penetration of the drug through the skin and extend the shelf life of the patches [14,15,16]. In addition, research has been carried out in terms of the mechanism of transdermal penetration of drug molecules with respect to their physicochemical properties, such as solubility (S), the presence of enantiomer (ET), the logarithm of the octanol-water partition coefficient (log P), molecular weight (MW), and melting point (MP) [17,18,19]. Scientists are also working on optimizing the film matrix to ibuprofen patch formulation to improve penetration of the drug from the adhesive matrix into the skin. For this purpose, the mixture of polyvinyl pyrrolidone (PVC) and hydroxypropyl methylcellulose (HPMC) was tested [2,20], or a novel micellar transdermal delivery of ibuprofen was prepared using the copolymeric excipient polyvinyl caprolactam-polyvinyl acetate-polyethylene glycol graft copolymer (Soluplus^®^) [21]. In the adhesives used in TDDS, we can also distinguish the synthesis of new copolymers, i.e., sustainable PSAs or those manufactured using modern and pro-ecological technologies, e.g., UV technology [21,22,23].

There are generally two types of transdermal patches: matrix (drug-in-adhesive) and reservoir. In the presented studies, matrix-based transdermal patches were developed which have an advantage over reservoir patches in terms of ease of use and manufacturability, the acceptable cost of the products, and the absence of dose dumping [24]. In addition, transdermal systems can modulate the drug release levels on site for long periods. Still, the absolute amount permeating the skin depends on the drug’s matrix [3]. Matrix transdermal patches are usually prepared using organic solvent-based pressure-sensitive adhesives (PSAs), such as acrylate copolymer, silicone, and polyisobutylene (PIB).

PSAs are soft polymeric materials that show permanent stickiness at room temperature and instantly adhere to surfaces when applying mild pressure. The adhesion performances can be regulated by the copolymer formulation, which will determine the glass transition temperature (Tg) and the shear modulus (G) [25]. The acrylic PSAs offer the highest balance of adhesion, cohesion, and excellent water resistance [26,27,28]. The performance requirements of medical PSAs based on acrylics are demanding as they must adhere well to varying skin types (both dry and moist), be removable without leaving adhesive residue or causing skin damage, and should not irritate the skin. Ideally, medical PSAs adhere strongly to the skin but can be easily removed with little or no trauma (adhesion properties) and without adhesive residues (cohesion properties) [26]. UV-crosslinkable acrylic PSA is becoming increasingly important due to the environmental hazards and medical applications associated with conventional crosslinkable solvent-borne PSAs and the performance shortcomings of PSAs based on aqueous systems [26].

In our previous studies, we have demonstrated that, with topically applied preparations, the vehicle’s composition can significantly impact the percutaneous penetration of the ibuprofen [23]. Hence, in this research work, we focused on showing the effect of new structural modifications of ibuprofen on the penetration of an acrylic drug-in-adhesive matrix type patch and their behavior in a PSA matrix with self-adhesive properties. The structural modifications of ibuprofen concern its salt’s formation by pairing the ibuprofen anion with organic cations, such as amino acid isopropyl esters. The structural modifications of ibuprofen used in the study were selected based on previous studies on the permeability of this type of compound through the skin [29]. The research results were analyzed in the context of the chemical structure of the drugs used because this appears to have a significant impact on the drug’s skin permeability and functional properties.

## 2. Results

### 2.1. Evaluation of the Self-Adhesive Properties of a Transdermal Patch Containing Various Active Substances

The pressure-sensitive adhesive (PSA) employed consists of a proprietary acrylate copolymer consisting of 2-ethylhexyl acrylate, hydroxyethyl acrylate, glycidyl methacrylate, and vinyl acetate. The acrylic PSA was thermally crosslinked to form a crosslinked polymer matrix. As ibuprofen and its structural modifications are present in the PSA during the crosslinking process, it is important to investigate their influence on the crosslinking process and self-adhesive properties (Table 1).

The obtained transdermal patches contained different active substances with different molecular weights. Therefore, the prepared patches had a different coat weight in the range of 17–40 g/m^2^. All patches were crosslinked under the same conditions. However, the number of solvents used to dissolve the active substances differed, influencing the solid weight content determined via gravimetry.

Shear strength (shear adhesion) reveals the resistance of a transdermal patch to tangential stresses and, therefore, the cohesion of the adhesive matrix [29]. It was shown that the addition of active substances significantly reduced the cohesion of the adhesive matrix. The average time taken for the patch to drop from the test surface was found to be from 1 to 11 min (Table 1). This corresponds to similar results obtained when testing different acrylic PSAs with pure ibuprofen.

Effective adhesion is an important feature of any transdermal patch, as the amount of drug delivered to the skin depends on the surface area of the patch adhering to it. Thus, a partially detached surface may reduce the amount of ibuprofen permeated across the skin. This is especially important when the patch has to be worn for a long period of time, e.g., 7 days. To overcome this problem, the patches should strongly adhere to the skin throughout this period. Therefore, the impact of the various active substances in an adhesive matrix on the adhesion of the resulting patch was assessed. The adhesion of the pure formulation showed (as expected) the highest adhesion (13.6 N). Upon the addition of various active substances, the adhesion decreased. It must be noted that the adhesion of all the obtained patches is still high and amounts to over 7.5 N. The exception is four patches containing Val-IBU, Ser-IBU, Lys-IBU, and IBUNa, whose adhesion was 6.77 N, 3.68 N, 3.17 N, and 0.08 N, respectively.

Similar results were obtained with the tack test, which assesses the effectiveness of transdermal patch adhesion by measuring the debonding force on applying light pressure for a short time. Again, the highest tack was the reference sample containing no active substances (14 N). In turn, the lowest values were recorded for Val-IBU (2.61 N), Thr-IBU (4.81 N), and Lys-IBU (2.03 N). The remaining tested adhesive films had tackiness at a similar level and range of 8.04–13.50 N.

### 2.2. Microscopy and Stability Assessment of Acrylate Transdermal Patches with Structurally Modified Ibuprofen

A short-term stability study (3 months) for the optimized formulation was also performed to assess the quality and estimate the resulting patch’s shelf-life. The samples were protected with siliconized film to simulate the conditions corresponding to the storage of patches and kept under constant temperature conditions. Furthermore, the drug crystals were also observed after 7 days in the case of patches not protected with siliconized foil, corresponding to the period of use of the patch on the skin. In addition, samples were evaluated for color change by organoleptic and crystallinity by microscopy. The research results were analyzed in the context of the chemical structure of the drugs used, i.e., their division, polarity, crystallization, and diffusion. The research shows that these types of variables have a significant impact on the permeability of the drug to the skin and its functional properties.

As a result of the conducted research, it has been shown that not all patches are stable during storage. The obtained patches are shown in Figure 1. As a result of their observation using organoleptic methods, in the case of most of the tested samples, no significant changes were found during the seasoning time. The exceptions are samples containing active substances in the form of IBUNa and Thr-IBU, which included white particles visible to the naked eye, as well as the sample from Ser-IBU, which changed color immediately after crosslinking from transparent to slightly orange, which may indicate its lack of resistance to the elevated temperature which was used during crosslinking of the adhesive matrix.

Some structural modifications of IBU dispersed in the adhesive matrix showed micrometric particle size which was recorded by microscopy. The observations were carried out on the day of receiving the patch, after 7 days simulating the conditions of the period of using the patch on the skin, and after 3 months, which was to reproduce the storage conditions of obtained patches in constant temperature (20 °C). Table 2 presents the crystallization of drugs quantitatively from the patch in the form of microscopic images, while Table 3 summarizes the sizes of the observed drug crystals. First, on receiving patches with new ibuprofen modifications, no significant differences in drug crystal sizes were observed compared to the reference patch containing unmodified ibuprofen (TP-IBU). The exception was the patch containing IBUNa which contained large drug crystals. This may be due to their poor solubility in solvents or the adhesive matrix in preparing adhesive compositions. After 7 days of observing patches not protected with siliconized foil, i.e., under conditions simulating their use, such as air access, small crystals are visible in all patches, regardless of the ibuprofen modification applied. Additionally, in this case, TP-IBUNa is distinguished by an increase in the size of the crystals during the seasoning. The final observation was on patches protected with siliconized foil, which were observed after 3 months. In this case, crystals of similar size were observed on the day of their preparation. Therefore, no drugs were synthesized from the patches, proving their stability during storage.

### 2.3. Microspectroscopy Analysis

A Fourier transform infrared (FTIR) with microscope attachment was used to study the physical and chemical interactions between drugs and additives. This method allowed for viewing the transdermal patch’s specified measurement site and simultaneously collecting the infrared (IR) spectrum. The adhesive matrix is an acrylate copolymer synthesized from various acrylate monomers (2-ethylhexyl acrylate, hydroxyethyl acryl glycidyl methacrylate, and vinyl acetate). Therefore, in the crosslinked adhesive matrix spectrum without any active substances (shown at the top in Figure 2A,B and Appendix A), strong vibration at 1734 cm^−1^ was contributed from −C=O groups. The spectrum of the adhesive matrix contains some bands also present in the spectrum of the active substance, i.e., corresponding to the bonds stretching vibrations between C-O atoms characteristic for ether (Ar-OR), ester (RCOOR’), and carboxylic acid (RCO-OH) groups, visible as bands at wavenumbers 1235, 1163, 1093, 1017, 966, and 722 cm^−1^, while bands at wavenumbers 1458 cm^−1^ and 870 cm^−1^ correspond to the C-C bonds in the aromatic ring present in the ibuprofen structure or the adhesive matrix. As with the adhesive matrix spectra, in the FTIR spectra of all ibuprofen derivatives, the characteristic sharp absorption band was observed in the range of 1752–1704 cm^−1^, attributed to the C=O stretching vibrations of the ester carbonyl group in the drug’s ester part. For ibuprofen, there was broadband at 1704 cm^−1^, which is characteristic of the C=O stretching vibrations of the carboxylic acid group. In ibuprofen derivatives, the FTIR spectra of this band were shifted, and additional bands assigned to the stretching vibrations v(COO^−^) were observed at approximately 1382–1380 cm^−1^, proving the ionic structure of ibuprofen derivatives. The other differences in the IR bands for active substances were negligible. Individual active substances have already been the subject of detailed spectroscopic studies published in previous scientific works by the team of Ossowicz-Rupniewska [30]. All characteristic absorption peaks present in the spectra of obtained transdermal patches (TP) containing active substances either coincide with the peaks of the adhesive matrix or the corresponding active substances and their shifts or new peaks are not observed. Therefore, it is concluded that transdermal patches with active substances such as ibuprofen and its structural modifications with amino acids show no chemical interaction between drugs and the adhesive matrix, demonstrated explicitly with the example of ibuprofen as a reference substance (Figure 2A). In the case of TP-IBUNa (Figure 2B), crystallization of the active substance was observed, which is also visible in the spectrum in the form of peaks characteristic of these active substances, such as an additional band present in the IBUNa spectrum at 3353 cm^−1^ corresponding to the presence of hydroxyl groups in the structure. This band is present both in the adhesive spectrum and in the crystallized drug’s place.

### 2.4. TG and DSC

The thermal stability of the obtained patches was tested using thermogravimetric analysis. The following parameters were determined: onset decomposition temperature and temperature corresponding to the weight loss of 50% (determined from TG curves) and maximum decomposition temperatures (determined from DTG curves). These properties were summarized in Table 4 and Appendix A. The adhesive layer of the patch without the addition of the active substance showed higher stability than those with the addition of ibuprofen, sodium ibuprofenate, or selected L-amino acid isopropyl ester ibuprofenates, which was consistent with our previous research [31]. The change in the stability of the adhesive layer is caused by the addition of an active substance with stability lower than that of the adhesive. The onset of degradation of the commercial adhesive DT54 is about 300 °C, while for medical patches, the value is from 44.9 to 141.6 °C, lower for TP-IBUNa and TP-PheIBU, respectively. The temperatures corresponding to a 50% weight loss of adhesives were from 333.0 (for TP-IBU) to 361.7 °C (for TP-ThrIBU). All values were similar regardless of whether the adhesive was with or without an active substance. The maximum decomposition temperatures ranged from 307.6 °C for TP-IBUNa to 392.5 °C for TP-LysIBU_2_. For most adhesives, the maximum decomposition temperatures were higher than temperatures corresponding to 50% weight loss.

The addition of the active substance also affected the glass transition temperature of the obtained patches, which was −45.98 °C for the patch without the API addition. In most cases, the glass transition temperature was comparable or slightly lower (by a maximum of 5.9° for the addition of ibuprofen). For three plasters with the addition of ibuprophenates of amino acid alkyl esters, i.e., lysine ([LysOiPr][IBU], [LysOiPr][IBU]_2_), and phenylalanine ([PheOiPr][IBU]), the obtained glass transition temperature was higher than the glass transition temperature of the plaster without the API addition. The results of these analyzes are presented in Table 4. The addition of the lysine derivative salt significantly increases the glass transition temperature of the patch. As can be seen, the higher the content of the amino acid part, the higher the glass transition temperature; the value for the monoibuprofenate salt was −9.57 °C. In comparison, the value for the bis(ibuprofenate) salt was −26.89 °C. All DSC curves are included in the Appendix A.

### 2.5. Contact Angle

Figure 3 shows photographs of water droplets on the surface of obtained (A–M) and commercial (N) transdermal patches. The commercial patch had the most hydrophilic surface, which is not so much due to the adhesive surface used but to the carrier material from which the patch was made.

The lower contact angle possesses higher hydrophilicity and the higher contact angle lower hydrophilicity. The contact angle was 94.2 ± 1.3° for the commercial patch and 134.7 ± 0.9° for the patch containing ibuprofen (IBU). In general, the addition of the active ingredient in the form of an isopropyl amino acid ester salt reduces the contact angle. This is most likely due to an increase in the hydrophilicity of the active substance through its modification. The exceptions are derivatives of non-polar amino acids such as alanine or valine. This suggests that such non-polar amino acids have an influence on the properties of the obtained surfaces. The highest contact angle, and thus, the most hydrophobic surface, was obtained for a patch containing unmodified ibuprofen, which is the most lipophilic (hydrophobic) compound.

### 2.6. Permeability, Release, and Accumulation in Skin Studies

The cumulative mass of the tested compounds in acceptor fluid, considering all time points, is presented in Figure 4. The content of IBU and its derivatives in the acceptor fluid collected during 24 h permeation is summarized in Table 5. The cumulative mass of the individual compounds, determined after 24 h of permeation, was as follows: TP-LysIBU_2_ > TP-LysIBU > TP-ThrIBU > TP-SerIBU > TP-PheIBU > TP-ValIBU > TP-AlaIBU > TP-AspIBU > TP-ProIBU > TP-GlyIBU > commercial product > TP-IBU and TP-IBUNa. Among the studied patches, TP-LysIBU_2_ permeated significantly higher than others; the cumulative amount of substance permeated during the 24 h study was 147.356 ± 14.215 µg IBU·cm^−2^ (Table 5, Figure 4). As can be seen, polar amino acid derivatives such as lysine, threonine, and serine usually exhibit better permeability. The exception is the aspartic acid derivative, for which a lower permeability was observed than non-polar amino acid derivatives such as phenylalanine, valine, or alanine, and is probably due to the presence of two esterified carboxyl groups. In this case, it was the introduction of two isopropyl chains. Interestingly, the best permeability of the active substance was obtained in the case of the bis(ibuprofenate) salt, in which both amino groups of lysine were protonated (*RS*)-2-(4-(2-methylpropyl)phenyl)propanoic acid.

The permeation rate was also determined as it measures the therapeutic effect obtained. Figure 5 shows the permeation rate for each time interval. In general, the highest permeation rate for all patches was obtained in the first hours of the measurement, especially in 0.5–2 h. It is clearly visible that all structural modifications of ibuprofen used for the tests increase the permeation rate of the active compound. Moreover, the patches with unmodified ibuprofen show a speed similar to the commercial product.

Permeability parameters were determined, such as flux (J_SS_, µg IBU/cm^2^·h), apparent permeability coefficient (K_P_·10^3^, cm/h), lag time (L_T_, min), skin diffusion coefficient (D, cm^2^/h), skin partition coefficient (K_m_), and percentage of drug penetrated after 24 h (Q_%24 h_). The obtained results are presented in Table 5. There were visible differences between the stream results obtained for various derivatives. The lowest concentration was obtained for sodium ibuprofenate (0.771 µg IBU/cm^2^·h) and the highest for TP-LysIBU_2_ (15.112 µg IBU/cm^2^·h). In the case of using ibuprofenates conjugated with the amino acid isopropyl ester cation, higher flows of active substance were obtained, except for TP-ValIBU. The permeability coefficient, a quantitative measure of the rate at which the molecule can penetrate the skin, was also determined. It is a complex parameter influenced by factors related to the drug, skin barrier, and interactions. For the tested patches, the permeability coefficient values ranged from 0.539·10^3^ cm/h for TP-IBUNa to 10.578·10^3^ cm/h for TP-LysIBU_2_. The lag time depended on the type of amino acid used for modification. As shown previously, the lag time does not correlate with the side chain’s polarity [31]. In this case, the influence of the adhesive-active substance interactions is visible, and the relationship is not as typical as in the case of studies on the permeability of aqueous solutions. The maximum lag time was observed for TP-IBUNa as 28.770 min, while the minimum for TP-ProIBU and TP-AspIBU was the same in both cases, 0.126 min.

The skin diffusion coefficient was approximately 0.017 and 3.969 cm/h for TP-IBUNa and TP-AspIBU, respectively. The equilibrium solubility of the drug in the stratum corneum in relation to its solubility in the vehicle was also determined. This parameter describes the ability of a drug to escape solution and travel to the outermost layers of the stratum corneum. Its values were generally higher than for ibuprofen (0.405·10^3^). They ranged from 1.468·10^3^ to 10.186·10^3^ for TP-GlyIBU and TP-LysIBU, respectively, except for TP-AspIBU and TP-ProIBU, for which the values were lower and amounted to 50 and 47, respectively. The percentage of the permeated compound was also determined. The highest value was obtained for TP-LysIBU_2_. The obtained results suggest that the tested compounds may promote skin permeability. The selection of an appropriate structural modification of an active compound should be based on many factors, such as molecular weight, solubility, and lipophilicity.

The cluster analysis graph shows the cumulative mass of the IBU and its derivatives measured over the entire 24-h permeation period (Figure 6). In this diagram, three distinct groups of patches characterized by similar permeation can be distinguished (circles A, B, and C), and a separate one with the highest permeability, TP-LysIBU_2_ (Figure 6). Generally, the IBU derivatives penetrated the patches to a greater extent than pure IBU, IBUNa, and commercial patches (Figure 7). The similarity between derivatives was found using the Mann–Whitney test, which showed a statistically significant difference between all derivatives and pure ibuprofen (*p* < 0.05) (Appendix A).

Figure 8 shows the mass of [IBU] and its derivatives accumulated in porcine skin after 24 h of penetration. All the compounds used accumulated in the skin. The lowest accumulation for derivative IBU values was obtained for TP-AlaIBU (30.602 ± 2.847 µg IBU/g skin), while the greatest accumulation in the skin was observed for TP-GlyIBU (65.237 ± 0.781 µg IBU/g skin) (Figure 8).

The release of the active compound from the medical patch was also determined. As shown in Figure 9, the highest amounts of API are released in the first 3 h, after which the release of API is inhibited. Therefore, the obtained patches do not limit the number of active ingredients released. In addition, the highest release rate (Figure 10) was observed in the first 10 min test. For medical patches used to relieve pain, these results make it possible to guarantee a high efficiency compared to commercially available preparations.

## 3. Materials and Methods

### 3.1. Materials

The commercial polyacrylate adhesive was prepared, i.e., DURO-TAK 378-2054 (DT54; viscosity: 1.46 Pa·s; SWC: 49.7%) drug-in-adhesives matrix type transdermal patch. This type of adhesive is an acrylate copolymer in a mixture of solvents such as propan-2-ol (10–20%), ethyl acetate (10–20%), n-heptane (1–5%), petroleum (1–5%), methylcyclohexene (1–5%), and toluene (1–3%). The copolymer was obtained by copolymerization of the following monomers: 2-ethylhexyl acrylate, acrylic acid, butyl acrylate, and vinyl acetate. The composition of the reaction mixture also included aluminum tris(2,4-pentanedionato-O,O′), pentane-2,4-dione, and azobisisobutyronitrile (AIBN).

The following component was also used to prepare transdermal patches as active substances: ibuprofen (99%) (IBU; as reference material) and ibuprofen sodium salts (≥98%, IBUNa) were obtained from Sigma Aldrich (Steinheim am Albuch, Germany). All the structural modifications of ibuprofen used in this research were described previously. They were obtained in accordance with the previously described method [31]. Amino acid isopropyl ester salts which showed the best permeability from saturated solutions in PBS (pH = 7.4) were selected for the tests. For this research, the following compounds were selected: [GlyOiPr][IBU], [AlaOiPr][IBU], [ValOiPr][IBU], [SerOiPr][IBU], [ThrOiPr][IBU], [(AspOiPr)_2_][IBU], [LysOiPr][IBU], [LysOiPr][IBU]_2_, [PheOiPr][IBU], and [ProOiPr][IBU] and are described in Table 6.

Other reagents used in the study used for the permeation tests were PBS buffer pH 7.4 (Merck, Darmstadt, Germany), high purity orthophosphoric acid (98%) obtained from Chempur (Piekary Śląskie, Poland), HPLC gradient grade acetonitrile (≥99.9%) and methanol (99.9%) provided by Sigma-Aldrich (Steinheim am Albuch, Germany), and anhydrous potassium dihydrogen phosphate (99%) (KH_2_PO_4_) provided by Merck (Darmstadt, Germany).

### 3.2. Preparation of Adhesive Films

The commercial acrylate copolymer constituted the adhesive matrix of the transdermal patches. The structural modifications to ibuprofen concern its salts’ formation by replacing the acid proton with the sodium cation or pairing the ibuprofen anion with organic cations, such as amino acids. First, adhesive compositions with unmodified ibuprofen were prepared, followed by chemical modifications of the structure of ibuprofen, such as ibuprofen in the form of its salt by replacing the acid proton with the sodium cation (IBUNa), or pairing the ibuprofen anion with organic cations, such as the amino acid isopropyl esters (Gly-IBU, Ala-IBU, Val-IBU, Ser-IBU, Thr-IBU, Asp-IBU, Lys-IBU, Lys-IBU_2_, Phe-IBU, and Pro-IBU). The weight ratio of adhesive matrix to active substance was calculated based on the adhesive characteristics, i.e., solids content, the basis weight depends on the applied thickness of the adhesive film, and the characteristics of the active substance, i.e., the molar mass and the initial assumption regarding the content of active substances in commercial products, i.e., 200 mg of the active substance (ibuprofen) for the surface of the adhesive film equal to 140 cm^2^. The adhesive compositions in this series were prepared by dissolving the active substance in ethyl acetate and then adding the mixture to the adhesive matrix. Next, the adhesive compositions were coated (250 µm) on a polyester film. The obtained polymer layers were thermally crosslinked in the next stage for 10 min at 110 °C. The resulting adhesive film layer was covered with siliconized release paper. Table 7 shows the weight ratio of the adhesive to the active substance used in the adhesive composition, and Table 8 shows the adhesive compositions for preparing transdermal patches.

### 3.3. Characterization and Performance Evaluation

The viscosity of the obtained adhesive compositions was determined with a Bohlin Visco 88 (Malvern Panalytical) viscometer. The measurement was carried out at a temperature of 20 °C using the C14 geometry at a speed of 20 rpm.

The solid weight content (SWC) was determined in accordance with ISO 3251 (140 °C, 30 min) using a moisture analyzer (Radwag MAX 60/NP).

The coat weight of the crosslinked adhesive films (after evaporation of the solvent) was measured with a circular punch 1009 with an area of 10 cm^2^ (Karl Schröder KG, Weinheim, Germany).

Thermal stability was determined through thermogravimetric analyses conducted using thermomicrobalance TG 209 F1 Libra by Netzsch. Samples of approximately 5 mg weight were heated at a rate of 10°/min in an oxidative atmosphere nitrogen (protective gas): 10 mL/min, air: 25 mL/min), and a temperature range of 25 to 1000 °C. Onset decomposition temperature was determined from the intersection of TG curve tangents. The temperatures corresponding to the fastest sample weight loss were determined from the first derivative of the TG curve (DTG curve).

A DSC analyzing technique with differential calorimeter Q-100 (TA Instruments, New Castle, DE, USA, 2004) was employed to determine the glass transition temperature of the adhesive. Samples were subjected to a heating cycle from −80 °C to +100 °C with a heating rate of 10°/min.

### 3.4. Self-Adhesive Properties

The following self-adhesive properties were tested for the coated, crosslinked adhesive: tack, adhesion, and cohesion at different temperatures. For this purpose, international standards AFERA and FINAT were used. In addition, the shear strength was tested in accordance with FINAT FTM 8, adhesion according to AFERA 4001, and tack according to AFERA 4015. Tests were carried out on a Zwick/Roell Z-25 testing machine. Our previous article has provided a detailed procedure for performing the self-adhesive test [23].

### 3.5. Microscopy and Stability Assessment of Acrylate Transdermal Patches

The prepared acrylate patches were stored at constant conditions (20 °C and 50% of humidity) and then observed for the occurrence of crystallization under an optical microscope (Delta Optical, with MC500-W3 5 MP camera). The camera attached to the microscope was used to capture images at magnifications of 10×. The drug crystals were observed for 7 days in the case of patches not protected with siliconized foil, corresponding to the period of use of the patch on the skin, and three months in the case of patches protected with siliconized foil simulating the conditions corresponding to the storage of the patches.

### 3.6. Infrared Microspectroscopy

Fourier transform infrared (FTIR) is a technique used to study the physical and chemical interactions between drugs and additives [2]. Each patch was measured using a Nicolet iS5 IR spectrophotometer (Thermo Scientific, Waltham, MA, USA) at wavenumbers 500–4000 cm^−1^ equipped with a SurveyIR™ Infrared Microspectroscopy accessory (Czitek, LLC, Danbury, CT, USA). Visual images were produced by a high-resolution, 5 mp CMOS color video camera, 2592 × 1944 maximum resolution, and a 1900 µm field of view. Image display, manipulation, capture, and documentation were made using the eSpot software, providing the interface to select the sampling and illumination modes. This method allows viewing and collecting infrared (IR) spectrum simultaneously. The samples were placed in the microsampling accessory of the spectrometer sample compartment and microscopic observation was made in transmission mode. After selecting the analysis place, the FTIR spectrum was recorded by collecting the data in ATR mode.

### 3.7. Contact Angle

The water contact angle was measured using a Dataphysics OCA 15EC goniometer (Filderstadt, Germany). Measurements were made on the adhesive surface of the patch. The contact angle analysis was performed 30 s after placing the drop on the patch. The transdermal patches from five different places were cut into 2 cm × 2 cm, measured, and the results were averaged.

### 3.8. In Vitro Skin Permeation Studies

The permeation experiments were carried out using Franz diffusion cells (Phoenix DB-6, ABL&E-JASCO, Wien, Austria) with diffusion areas of 1 cm^2^. The acceptor chamber equipped with a magnetic stirring bar was filled with 10 mL PBS solution (pH 7.4). In each diffusion unit, a constant temperature of 37.0 ± 0.5 °C and speed of stirring was maintained. Porcine skin was used in the experiment since it is characterized by similar permeability to human skin [32,33]. The fresh abdominal porcine skin from the local slaughterhouse was washed in PBS buffer pH 7.4 several times. The thickness of the 0.5 mm skin was dermatomed and then divided into 2 cm × 2 cm pieces. The skin samples were wrapped in aluminum foil and stored in a freezer at −20 °C until use. The storage time was not longer than three months to maintain the skin barrier properties [34]. Before use, the skin samples were slowly thawed at room temperature for 30 min and hydrated with PBS pH 7.4 [33,34,35]. Undamaged skin pieces with an even thickness were chosen for experiments and mounted on the donor chamber.

The integrity of the skin was examined by measuring its impedance analogously, as described previously [36,37]. Only skin samples of impedance >3 kΩ were applied, which is similar to the value of the electrical resistance of human skin [37]. The patches were applied to the skin. Accordingly, as in the before-described methods [23], the experiment was carried out for 24 h, and the samples were collected at predefined times.

The release tests were performed in the same procedure as the permeation test, with the difference that the membrane was used. The release was performed according to the modified procedure of Song et al. [38] using the Franz-type receiver cell (Phoenix DB-6, ABL&E-JASCO, Wien, Austria) with diffusion areas of 1 cm^2^. The diffusion areas were covered with a dialysis tubing cellulose membrane (D 9777-100FT, Sigma Aldrich), and the prepared patch was mounted over the membrane. The acceptor cell was filled with 8 mL of pH 7.4 PBS. The release medium was maintained at 37 °C and the experiment was carried out for 24 h. The samples were reported after 10, 20, 30, 40, 50, 60, 70, 80, 100, 120 min, and 24 h of stirring. After this time, the acceptor fluid (0.4 mL) aliquots were withdrawn and refilled with fresh buffer at the same pH. HPLC measured the concentration of the compounds in the acceptor fluid.

The accumulation of the tested substance in the skin after penetration was determined using the methods described previously [11,24,31,34]. The supernatant was collected and analyzed by the HPLC method. Accumulation of IBU in the skin was calculated by dividing the amount of the drug remaining in the skin by the mass of the skin sample and is expressed as the mass of ibuprofen per mass of the skin (μg/g). A liquid chromatography system (Knauer, Berlin, Germany) assessed the concentration of IBU and its derivatives in the acceptor fluid in permeation and release tests and accumulation in the skin.

The HPLC system consisted of a model 2600 UV detector, Smartline model 1050 pump, and Smartline model 3950 autosampler with ClarityChrom 2009 software (Knauer, Berlin, Germany). The detector was operated at 264 nm. A 125 × 4 mm chromatographic column filled with Hyperisil ODS (C18), particle size 5 µm, was used. The mobile phase was 0.02 M potassium dihydrogen phosphate–acetonitrile (60/40 *v*/*v*) with a flow rate of 1 mL min^−1^. The column temperature was set at 25 °C and the injection volume was 20 μL.

HPLC measured the IBU and its derivative concentrations in the acceptor phase. The cumulative mass (µg IBU/cm^2^) was calculated based on this concentration. The flux (in µg IBU/cm^2^·h^1^) through the pigskin into acceptor fluid was determined as the slope of the plot of cumulative mass in the acceptor fluid versus time.

### 3.9. Statistical Analysis

Results are presented as the mean ± standard deviation (SD). A one-way analysis of variance (ANOVA) was used in the study. In the case of the cumulative mass, the significance of differences between individual groups was evaluated with Tukey’s test (α < 0.05). A cluster analysis was carried out to determine similarities between all patches tested, taking into account all time points in which the acceptor fluid was collected, presenting groups of compounds with a similar permeation. The Mann–Whitney test estimated significant differences in the cumulative mass between all analyzed patches, taking into account all time points during the 24 h permeation, which were then assessed by the Mann–Whitney test, where each derivative was compared to the control. Statistical calculations were done using Statistica 13 PL software (StatSoft, Kraków, Polska).

## 4. Conclusions

The structural modification of drugs such as ibuprofen salts obtained by pairing the ibuprofen anion with organic cations, such as amino acid isopropyl esters, may be an effective way to penetrate the drug through the skin, use its accumulation in the skin, and release treatments from the solid form of the drug. This paper presents twelve transdermal patches with newly developed drug modifications whose adhesive carrier is an acrylate copolymer. The developed TPs are, in most cases, characterized by good self-adhesive properties, stability during use and storage, and do not have interactions between the active substance and the adhesive. Results also indicate that the obtained ibuprofen patches show similar permeability to commercial patches compared to those with structural modifications of ibuprofen. However, these modified patches show an increased drug permeability of 2.3 to even 6.4 times greater than unmodified ibuprofen.

## Figures and Tables

**Figure 1 ijms-23-07752-f001:**
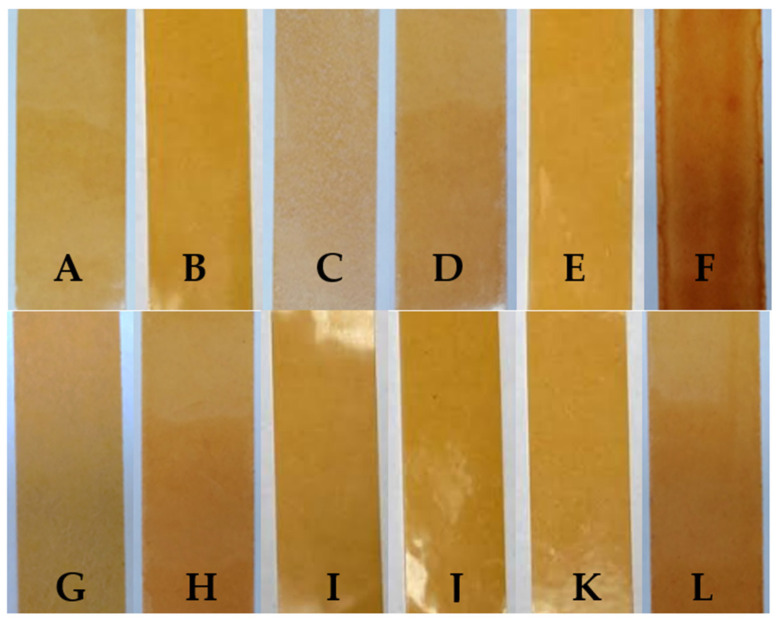
The acrylate-based transdermal patches are protected with yellow siliconized foil: (**A**)—TP-IBU; (**B**)—TP-IBUNa; (**C**)—TP-GlyIBU; (**D**)—TP-AlaIBU; (**E**)—TP-ValIBU; (**F**)—TP-SerIBU; (**G**)—TP-ThrIBU; (**H**)—TP-AspIBU; (**I**)—TP-LysIBU; (**J**)—TP-LysIBU2; (**K**)—TP-PheIBU; and (**L**)—TP-ProIBU.

**Figure 2 ijms-23-07752-f002:**
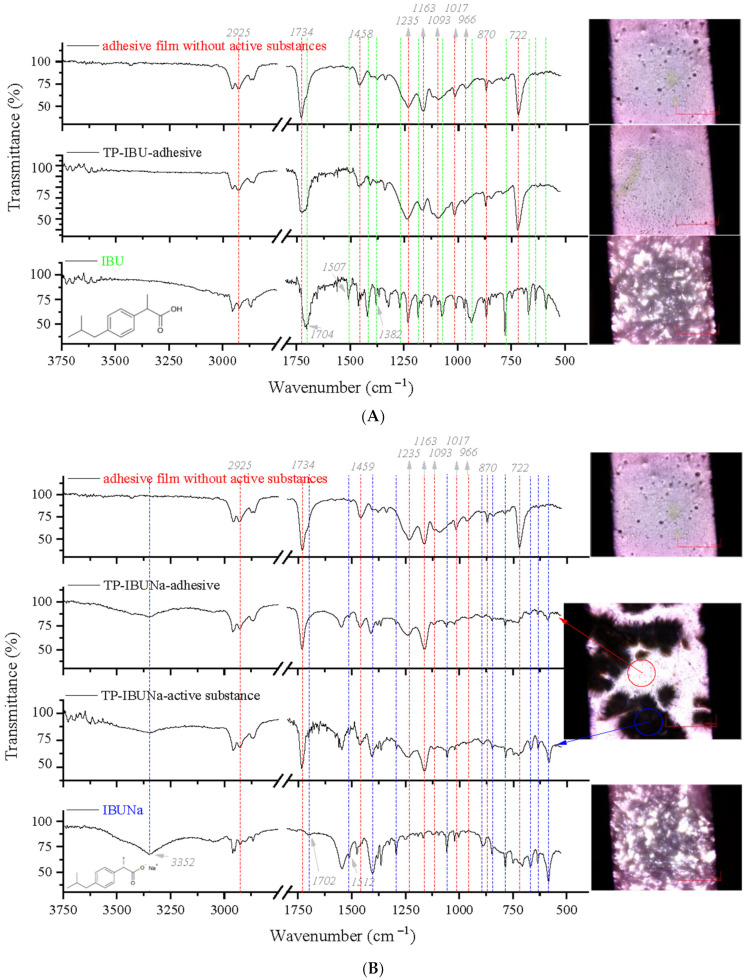
Microspectroscopy analysis of the transdermal patch (TP): (**A**) with the ibuprofen (IBU); (**B**) with IBUNa.

**Figure 3 ijms-23-07752-f003:**
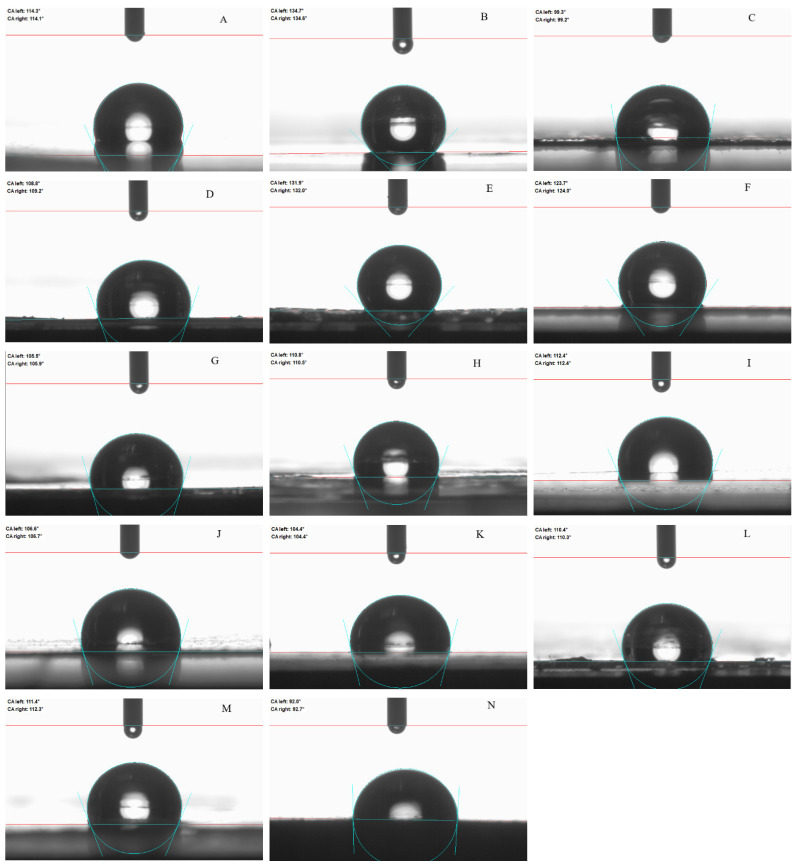
Water contact angle of the transdermal patch (TP): **A**) without active substance (DT54); (**B**) with the ibuprofen (IBU); (**C**) IBUNa; (**D**) GlyIBU; (**E**) AlaIBU; (**F**) ValIBU; (**G**) SerIBU; (**H**) ThrIBU; (**I**) AspIBU; (**J**) LysIBU; (**K**) LysIBU_2_; (**L**) PheIBU; (**M**) ProIBU; and (**N**) commercial product.

**Figure 4 ijms-23-07752-f004:**
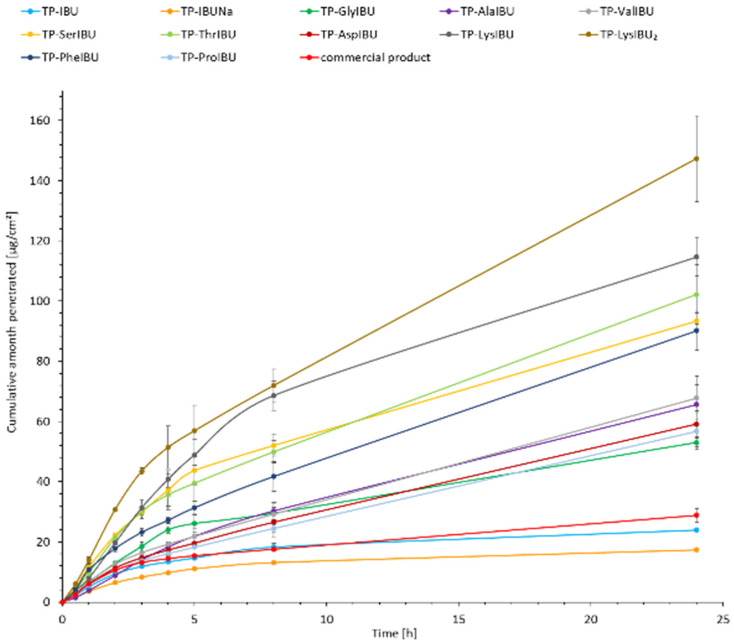
Ibuprofen and amino acid isopropyl ester ibuprofenates permeation profiles. Values are the means with standard deviation; *n* = 3.

**Figure 5 ijms-23-07752-f005:**
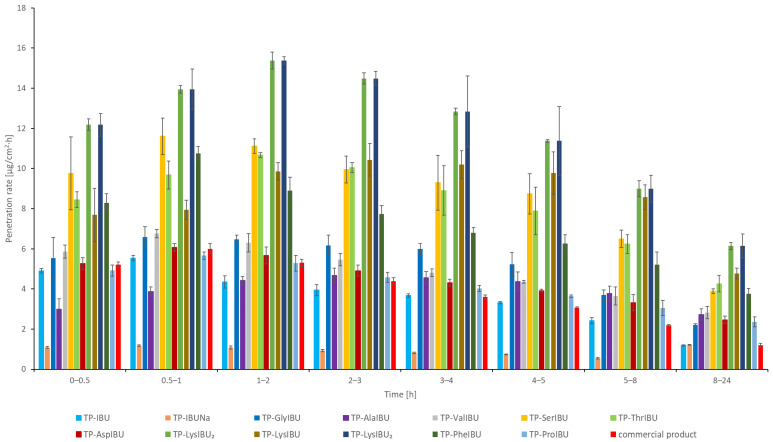
The permeation rate of ibuprofen and amino acid isopropyl ester ibuprofenates during the 24 h permeation; α = 0.05 (mean ± SD, *n* = 3).

**Figure 6 ijms-23-07752-f006:**
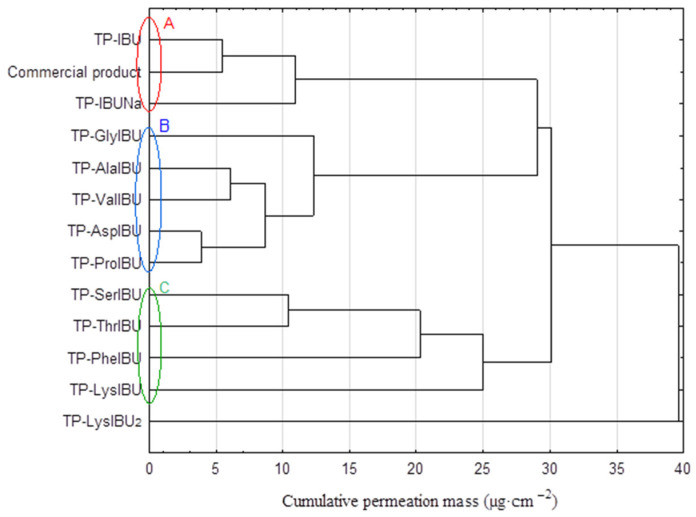
Cluster analysis graph for the mean accumulated mass of IBU and its derivatives during the entire 24 h study. Compounds with similar values are marked with a different color circles (A, B, C).

**Figure 7 ijms-23-07752-f007:**
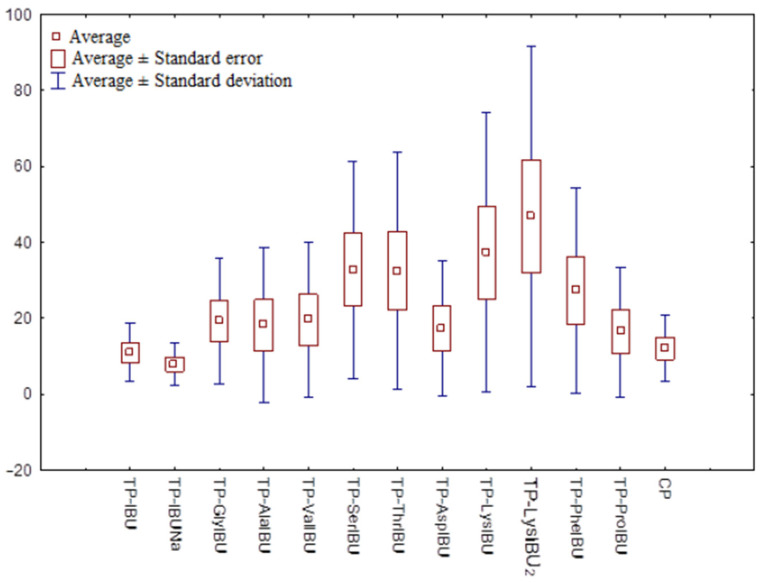
The box plot of cumulative mass for IBU and its derivatives after 24 h permeation.

**Figure 8 ijms-23-07752-f008:**
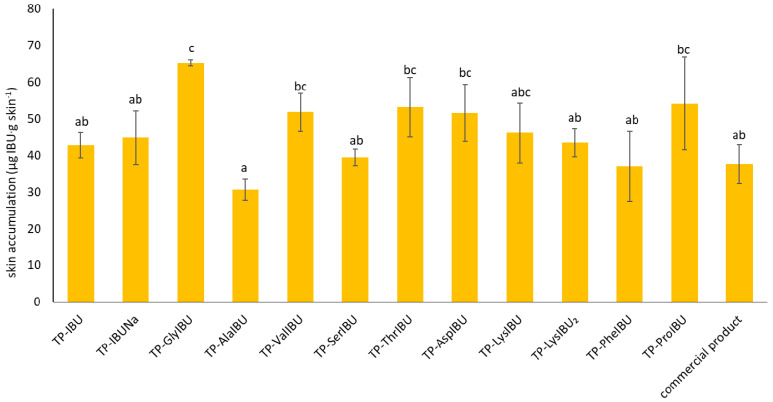
Significant differences in the cumulative mass between all analyzed compounds, taking into account all time points during the entire 24 h permeation, as estimated by Tuckey’s test, different letters—important differences between obtained transdermal patches and commercial product.

**Figure 9 ijms-23-07752-f009:**
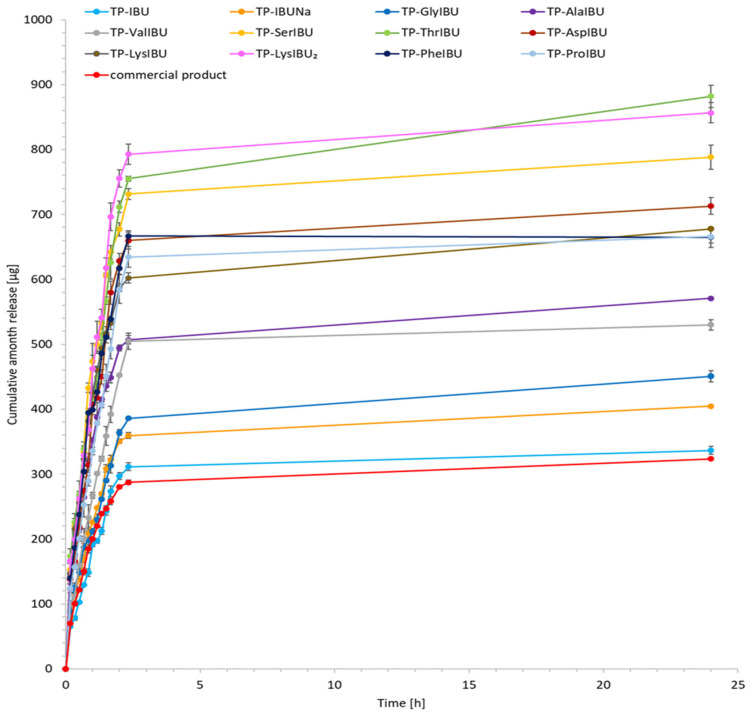
Time course of the cumulative mass of IBU and its derivatives during the 24 h release (mean ± SD, *n* = 3).

**Figure 10 ijms-23-07752-f010:**
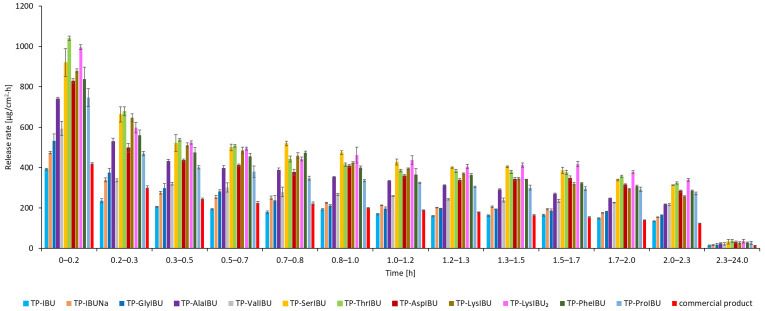
The release rate of ibuprofen and amino acid isopropyl ester ibuprofenates during the 24 h release; α = 0.05 (mean ± SD, *n* = 3).

**Table 1 ijms-23-07752-t001:** Self-adhesive properties of the transdermal patch containing various active substances.

Sample Code	Coat Weight (g/m^2^)	SWC (%)	Shear Strength	Adhesion (N/25 mm)	Tack (N)
DT54	32	98	>72 h	13.60	14.00
TP-IBU	40	97	10 min/c.f.	11.90/c.f.	13.50
TP-IBUNa	28	97	1 min 26 s/c.f.	0.08/c.f.	0.15/c.f.
TP-GlyIBU	28	93	3 min 11 s/c.f.	10.14/c.f.	8.60
TP-AlaIBU	27	92	1 min 42 s/c.f.	11.48/c.f.	11.18
TP-ValIBU	17	96	2 min 16 s/c.f.	6.77/c.f.	2.61
TP-SerIBU	23	92	1 min/c.f.	3.68/c.f.	8.08/c.f.
TP-ThrIBU	39	91	1 min 18 s/c.f.	9.16/c.f.	4.81/c.f.
TP-AspIBU	38	90	48 s/c.f.	7.78/c.f.	9.68/c.f.
TP-LysIBU	25	94	17 h 18 min	3.17	2.03
TP-LysIBU_2_	35	93	6 min 8 s/c.f.	13.32/c.f.	8.72
TP-PheIBU	22	90	11 min/c.f.	11.26/c.f.	8.04
TP-ProIBU	35	94	1 min 10 s/c.f.	7.57/c.f.	11.55

SWC—Solid weight content determined via gravimetry; c.f.—cohesive failure.

**Table 2 ijms-23-07752-t002:** Microscopic observation of patch samples containing various structural modifications of ibuprofen during the seasoning time.

Sample Code	First Day of Observation	Observation after 7 Days (Patches Not Protected with Siliconized Foil)	Observation after 3 Months (Patches Protected with Siliconized Foil)
TP-IBU	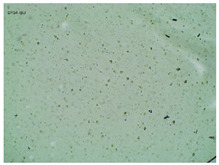	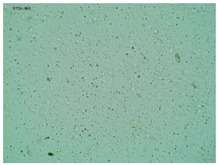	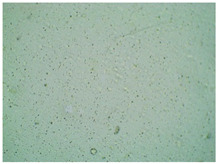
TP-IBUNa	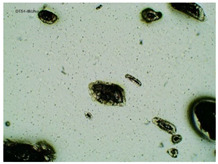	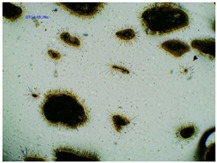	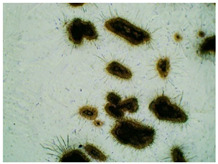
TP-GlyIBU	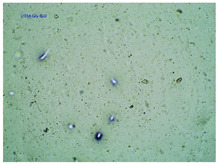	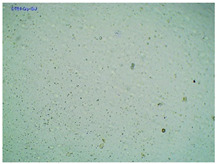	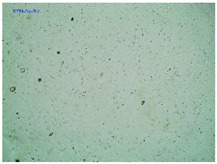
TP-AlaIBU	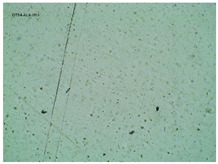	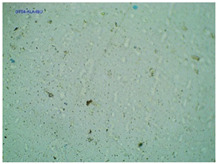	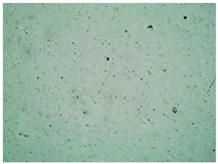
TP-ValIBU	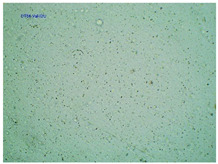	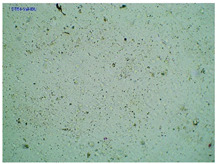	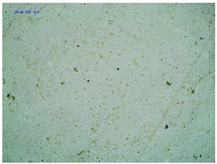
TP-SerIBU	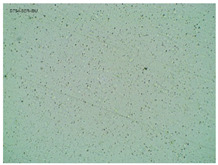	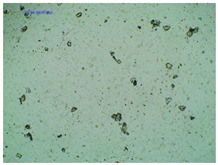	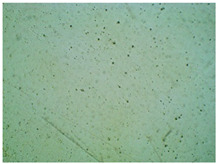
TP-ThrIBU	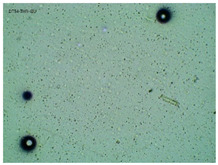	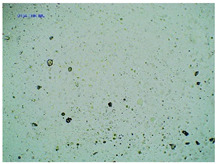	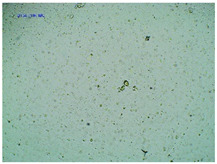
TP-AspIBU	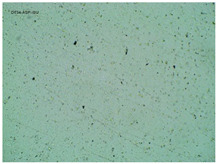	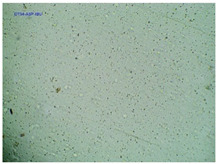	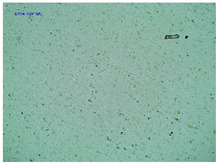
TP-LysIBU	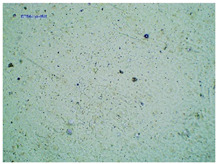	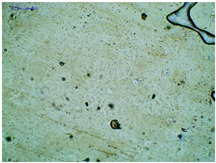	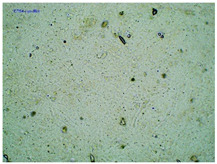
TP-LysIBU_2_	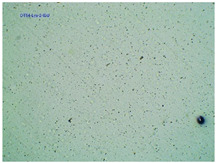	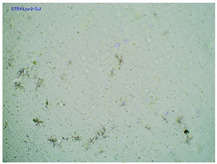	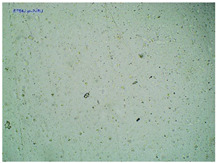
TP-PheIBU	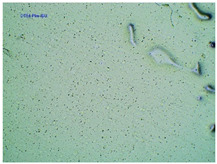	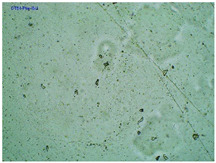	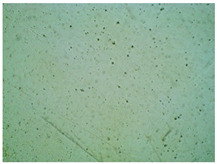
TP-Pro-IBU	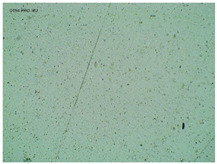	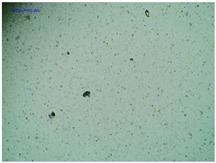	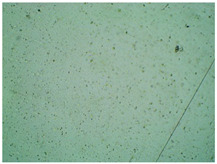

**Table 3 ijms-23-07752-t003:** The size of drug crystals determined by microscopy.

Sample Code	First Day of Observation	Patches Not Protected with Siliconized Foil—Observation after 7 Days	Patches Protected with Siliconized Foil—Observation after 3 Months
TP-IBU	26 ± 6 µm	51 ± 4 µm	19 ± 9 µm
TP-IBUNa	168 ± 64 µm	263 ± 112 µm	231 ± 13 µm
TP-GlyIBU	28 ± 5 µm	38 ± 12 µm	27 ± 4 µm
TP-AlaIBU	32 ± 5 µm	52 ± 9 µm	24 ± 9 µm
TP-ValIBU	24 ± 9 µm	28 ± 7 µm	24 ± 3 µm
TP-SerIBU	13 ± 6 µm	57 ± 14 µm	13 ± 5 µm
TP-ThrIBU	32 ± 9 µm	36 ± 4 µm	29 ± 6 µm
TP-AspIBU	26 ± 9 µm	36 ± 6 µm	23 ± 6 µm
TP-LysIBU	23 ± 3 µm	45 ± 14 µm	36 ± 4 µm
TP-LysIBU_2_	26 ± 2 µm	67 ± 9 µm	39 ± 4 µm
TP-PheIBU	20 ± 7 µm	41 ± 9 µm	23 ± 5 µm
TP-ProIBU	17 ± 13 µm	38 ± 6 µm	15 ± 2 µm

**Table 4 ijms-23-07752-t004:** Glass transitions and the thermal stability of acrylic PSAs and PSAs with ibuprofen (IBU) and its structural modifications.

Sample Code	T_g_ (°C)	T_IDT_ (°C)	T_d_^50%^ (°C)	T_MDT_ (°C)
DT54	−45.98	301.1	357.6	347.1
TP-IBU	−51.88	165.8	333.0	348.6
TP-IBUNa	nd	256.2	348.1	307.6
TP-GlyIBU	−46.14	166.6	356.2	378.4
TP-AlaIBU	−51.06	182.0	351.4	361.3
TP-ValIBU	−48.35	168.3	350.1	361.6
TP-SerIBU	−47.01	185.4	346.2	371.2
TP-ThrIBU	−50.47	181.6	361.7	382.0
TP-AspIBU	−51.23	180.9	338.8	392.0
TP-LysIBU	−9.57	164.2	351.8	390.3
TP-LysIBU_2_	−26.89	184.3	333.8	392.5
TP-PheIBU	−43.64	159.5	338.1	360.7
TP-ProIBU	−51.34	187.1	354.8	381.9

T_g_—glass transitions, T_IDT_—onset decomposition temperature, T_d_^50%^—50% weight loss temperature, T_MDT_—maximum decomposition temperature, and nd—not detected.

**Table 5 ijms-23-07752-t005:** Skin permeation parameters for ibuprofen and amino acid isopropyl ester ibuprofenates; different letters indicate significant differences between the tested compounds, α = 0.05, mean ± SD, and *n* = 3. The statistically significant difference was estimated by ANOVA using Tuckey’s test.

Sample Code	Cumulative Permeation Mass, µg IBU/cm^2^	J_SS_, µg IBU/cm^2^·h	K_P_·10^3^, cm/h	L_T_, min	D, cm^2^/h	K_m_·10^3^	Q%_24 h_
TP-IBU	23.979 ± 0.547 ^a^	4.702	3.291	1.231	0.406	0.405	1.679
TP-IBUNa	17.378 ± 1.408 ^a^	0.771	0.539	28.770	0.017	1.552	0.406
TP-GlyIBU	53.019 ± 1.519 ^b^	6.005	4.204	3.492	0.143	1.468	3.711
TP-AlaIBU	65.601 ± 6.542 ^b^	4.852	3.396	11.003	0.045	3.736	4.592
TP-ValIBU	67.741 ± 7.244 ^b^	4.620	3.234	24.413	0.020	7.894	4.742
TP-SerIBU	93.343 ± 2.673 ^c^	11.431	8.002	2.271	0.222	1.817	6.534
TP-ThrIBU	102.211± 9.860 ^cd^	10.534	7.374	3.932	0.128	2.899	7.155
TP-AspIBU	59.143 ± 4.307 ^b^	5.716	4.001	0.126	3.969	0.050	4.140
TP-LysIBU	114.653 ± 6.375 ^d^	11.157	7.810	13.042	0.038	10.186	8.026
TP-LysIBU_2_	147.356 ± 14.215 ^e^	15.112	10.578	0.655	0.127	4.159	10.315
TP-PheIBU	90.132 ± 6.563 ^c^	8.821	6.174	4.193	0.119	2.589	6.309
TP-ProIBU	56.765 ± 6.071 ^b^	5.328	3.729	0.126	3.964	0.047	3.974
Commercial product	28.817 ± 2.158 ^a^	5.226	3.658	3.487	0.143	1.276	2.017

J_SS_—steady-state flux; K_P_—permeability coefficient; L_T_—Lag time; D—diffusion coefficient in the skin; K_m_—skin partition coefficient; and Q_%24 h_—percent drug permeated after 24 h.

**Table 6 ijms-23-07752-t006:** Characteristics of the used active substances.

Symbol	Name	M_mol_ (g/mol)	Chemical Structure
IBU	(*RS*)-ibuprofen	206.28	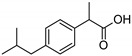
IBUNa	ibuprofen sodium salts	228.26	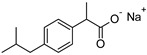
Gly-IBU	[GlyOiPr][IBU]	323.43	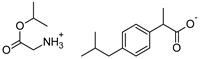
Ala-IBU	[AlaOiPr][IBU]	337.45	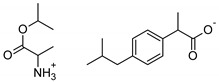
Val-IBU	[ValOiPr][IBU]	365.51	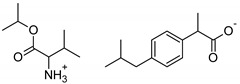
Ser-IBU	[SerOiPr][IBU]	353.45	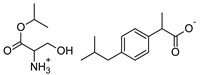
Thr-IBU	[ThrOiPr][IBU]	367.48	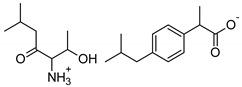
Asp-IBU	[(AspOiPr)_2_][IBU]	423.54	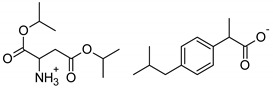
Lys-IBU	[LysOiPr][IBU]	394.55	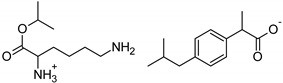
Lys-IBU_2_	[LysOiPr][IBU]_2_	600.83	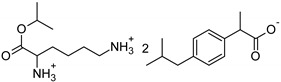
Phe-IBU	[PheOiPr][IBU]	413.55	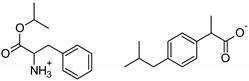
Pro-IBU	[ProOiPr][IBU]	363.49	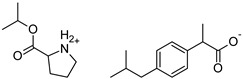

**Table 7 ijms-23-07752-t007:** The weight ratio of the adhesive to the active substance in adhesive compositions.

Sample Code of the Adhesive Composition Containing the Active Substance	PSA Characteristics				Active Substance Characteristics		The Weight Ratio of the Adhesive to the Active Substance	
	Symbol	SWC (%)	l (µm)	Coat Weight (g/m^2^)	Symbol	M_mol_ (g/mol)	PSA (g)	Active Substance (g) ^*^
TP-IBU	DT54	49.7	250	32	IBU	206.28	0.901	0.200
TP-IBUNa					IBUNa	228.26		0.221
TP-GlyIBU					Gly-IBU	323.43		0.314
TP-AlaIBU					Ala-IBU	337.45		0.327
TP-ValIBU					Val-IBU	365.51		0.354
TP-Ser-IBU					Ser-IBU	353.45		0.343
TP-ThrIBU					Thr-IBU	367.48		0.356
TP-AspIBU					Asp-IBU	423.54		0.411
TP-LysIBU					Lys-IBU	394.55		0.383
TP-LysIBU_2_					Lys-IBU_2_	600.83		0.583
TP-PheIBU					Phe-IBU	413.55		0.401
TP-ProIBU					Pro-IBU	363.49		0.352

* assumption: 200 mg of active substance/140 cm^2^; SWC—Solid weight content determined via gravimetry; (1) l—the thickness of the adhesive film without active substance (before crosslinking); (2) Coat weight—coat weight without active substance (after evaporation of the solvent and crosslinking).

**Table 8 ijms-23-07752-t008:** Adhesive compositions for the preparation of transdermal patches.

Sample Code of the Adhesive Composition Containing the Active Substance	PSA		Active Substance		Solvent	
	Symbol	Weight (g)	Symbol	Weight (g)	Symbol	Weight (g)
TP-IBU	DT54	4.51	IBU	1.00	OE	1.00
TP-IBUNa			IBUNa	1.11		2.00
TP-GlyIBU			Gly-IBU	1.57		4.00
TP-AlaIBU			Ala-IBU	1.64		2.00
TP-ValIBU			Gly-IBU	1.77		8.00
TP-Ser-IBU			Ser-IBU	1.71		6.00
TP-ThrIBU			Thr-IBU	1.78		2.00
TP-AspIBU			Asp-IBU	2.05		1.00
TP-LysIBU			Lys-IBU	1.91		6.00
TP-LysIBU_2_			Lys-IBU_2_	2.91		3.00
TP-PheIBU			Phe-IBU	2.01		7.00
TP-ProIBU			Pro-IBU	1.76		2.00

## Data Availability

The data presented in this study are available on request from the corresponding author.

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
