# Peer review of "Evaluation of the Structural Modification of Ibuprofen on the Penetration Release of Ibuprofen from a Drug-in-Adhesive Matrix Type Transdermal Patch"

_ijms, 2022, doi:10.3390/ijms23147752_

Round 1
Reviewer 1 Report
The manuscript studied the ibuprofen usage as a transdermal local analgesic. The authors used various IBU active materials of Ibuprofen sodium salt, [GlyOiPr][IBU], [AlaOiPr][IBU], [ValOiPr][IBU], [SerOiPr][IBU], 24 [ThrOiPr][IBU], [(AspOiPr)2][IBU], [LysOiPr][IBU], [LysOiPr][IBU]2, [PheOiPr][IBU], and 25 [ProOiPr][IBU]. The data shows the stable and effective using of these materials with high permeability. The manuscript was written well.
Please address the following comments:
1- The abstract section was written generally; it is better to redesign the abstract by focusing on the merit of the figure.
2- For microscopy analysis, it is better to add the Raman shift to configure the interaction between patch materials and IBU. It is better to expand this discussion to outline all materials. In addition, it is better to move some data from Figure 2 into supporting information, I think Figures 2 A and B are enough.
3- It is highly recommended to address the presence and structure of the material on the patch surface using FE-SEM.
in addition, it is better to add the EDX-SEM to support the presence of various materials at the patch's surface.
4- It is highly recommended to show the contact angle of the loaded materials at the patch's surface before and after adding the IBU.
Author Response
Dear reviewer,
we would like to thank you very much for taking the time to give Your opinion on our publication. Please find attached responses to Your objections and comments. We hope the article will meet your requirements.
With best regards,
Authors

Reviewer 2 Report
Please precise the temperature condition for the short term stability (line163) is it 20°C (line 191) what about the humidity ?
Table 1 if possible duplicate the title page 6 to the page 7 (table 2 ?? where is table 2 ?). If possible indicate a scale measurement in the picture. data may by available (eg table 3)
Fig 8 there is no dat from 2,3 H to 24 H. In fig 3 there is a point at 8H ??
Please add legend axe X into fig 4 and fig 9
Please change the chapter order, chapter 2 =materiel and methods and chapter 3 Results. Please pay attention to paragraph number (page 24 3.3 , page 25 2.5 2,6...... etc
Author Response

(The authors gave the same response as above.)

Round 2
Reviewer 1 Report
The authors have addressed most of the comments and modified the manuscript. So, I recommend accepting the manuscript in its current form